# Cancer Vulnerabilities Through Targeting the ATR/Chk1 and ATM/Chk2 Axes in the Context of DNA Damage

**DOI:** 10.3390/cells14100748

**Published:** 2025-05-20

**Authors:** Anell Fernandez, Maider Artola, Sergio Leon, Nerea Otegui, Aroa Jimeno, Diego Serrano, Alfonso Calvo

**Affiliations:** 1Program in Solid Tumors, CIMA, Cancer Center Clinica Universidad de Navarra (CCUN), University of Navarra, Avenida de Pio XII, 55, 31008 Pamplona, Spain; afernandezl@alumni.unav.es (A.F.); sleon1@unav.es (S.L.); notegui@unav.es (N.O.); dserrano@unav.es (D.S.); 2Department of Pathology, Anatomy and Physiology, School of Medicine, University of Navarra, 31008 Pamplona, Spain; martolalarr@unav.es (M.A.); ajimenojuri@alumni.unav.es (A.J.); 3CIBERONC, ISCIII, 28029 Madrid, Spain; 4IDISNA, 31008 Pamplona, Spain

**Keywords:** ATR, Chk1, ATM, Chk2, DNA damage, synthetic lethality, immunotherapy

## Abstract

Eliciting DNA damage in tumor cells continues to be one of the most successful strategies against cancer. This is the case for classical chemotherapy drugs and radiotherapy. In the modern era of personalized medicine, this strategy tries to identify specific vulnerabilities found in each patient’s tumor, to inflict DNA damage in certain cell contexts that end up in massive cancer cell death. Cells rely on multiple DNA repair pathways to fix DNA damage, but cancer cells frequently exhibit defects in these pathways, many times being tolerant to the damage. Key vulnerabilities, such as *BRCA1/BRCA2* mutations, have been exploited with PARP inhibitors, leveraging synthetic lethality to selectively kill tumor cells and improving patients’ survival. In the DNA damage response (DDR) network, kinases ATM, ATR, Chk1, and Chk2 coordinate DNA repair, cell cycle arrest, and apoptosis. Inhibiting these proteins enhances tumor sensitivity to DNA-damaging therapies, especially in DDR-deficient cancers. Several small-molecule inhibitors targeting ATM/Chk2 or ATR/Chk1 are currently being tested in preclinical and/or clinical settings, showing promise in cancer models and patients. Additionally, pharmacological blockade of ATM/Chk2 and ATR/Chk1 axes enhances the effects of immunotherapy by increasing tumor immunogenicity, promoting T-cell infiltration and activating immune responses. Combining ATM/Chk2- or ATR/Chk1-targeting drugs with conventional chemotherapy, radiotherapy or immune checkpoint inhibitors offers a compelling strategy to improve treatment efficacy, overcome resistance, and enhance patients’ survival in modern oncology.

## 1. Introduction

Chemotherapy and radiotherapy are cornerstones for cancer treatment. Many chemotherapy drugs and radiotherapy act through damaging the DNA and causing cell death [1]. Irradiation induces DNA breaks, especially double-strand breaks, which are difficult for cells to repair [2]. Chemotherapy with DNA damaging drugs includes alkylating agents, platinum-based compounds, and topoisomerase inhibitors. These drugs exert their effects by interfering with DNA replication and transcription, ultimately leading to cell death. Cancer cells, which divide more rapidly and frequently than most normal cells, are especially vulnerable to this type of damage, making DNA-damaging chemotherapy an effective therapeutic strategy. Radiotherapy and chemotherapy are many times combined to produce a synergistic anticancer effect in tumors [3]. However, these therapies are also associated with side effects, such as hair loss, bone marrow suppression, and gastrointestinal toxicity. In the era of precision medicine, new approaches to damage the DNA and block DNA repair mechanisms, based on particular alterations of each tumor, are being intensely investigated. These therapies, many of which target DNA repair pathways, are expected to be highly effective with fewer toxic side effects. Targeting specific proteins involved in the DNA repair pathway also presents a promising strategy for sensitizing tumor cells to chemotherapy and radiotherapy [4].

DNA is continuously exposed to various internal and external insults that can lead to mutations, which, if left unrepaired, may contribute to carcinogenesis. To maintain genomic integrity, cells use several DNA repair mechanisms, each specialized in correcting different types of damage. The primary DNA repair pathways include base excision repair (BER), nucleotide excision repair (NER), mismatch repair (MMR), homologous recombination (HR), and non-homologous end joining (NHEJ) [5]. BER corrects small, non-helix-distorting base alterations caused by oxidation or alkylation, while NER removes bulky helix-distorting lesions, including those induced by ultraviolet (UV) light [5]. MMR solves replication errors, such as base mismatches and insertion-deletion loops. Double-strand breaks (DSBs), which are particularly harmful for cells, are repaired by HR (an error-free process using the sister chromatid as a template) or NHEJ, which ligates broken ends directly and is more error-prone [5].

These repair mechanisms are frequently altered in cancer cells, which are able to adapt by tolerating DNA damage and accumulating mutations, instead of dying. Mutations or deficiencies in proteins of these pathways are common. For instance, *BRCA1* and *BRCA2* mutations impair HR, increasing susceptibility to DNA damaging drugs in breast and ovarian cancers. This vulnerability has been exploited therapeutically using PARP inhibitors (e.g., olaparib). PARP plays a role in single-strand DNA break repair, and its inhibition in *BRCA*-deficient cells leads to the so-called synthetic lethality: a synergistic effect produced by blockade of two or more pathways related to a specific cellular process, leading to massive cell death [6]. Exploiting synthetic lethality enables selective targeting of tumor cells while sparing normal tissue. Ongoing research is focused on identifying novel repair pathway dependencies and developing selective inhibitors to enhance the effectiveness and specificity of cancer therapies [7].

The DNA damage response (DDR) pathway plays a crucial role in maintaining genomic stability, and its dysregulation is a hallmark of cancer. Among the central DDR proteins are the kinases ATM (ataxia-telangiectasia mutated), ATR (ATM and Rad3-related), Chk1 (Checkpoint Kinase 1), and Chk2 (Checkpoint Kinase 2), which coordinate cell cycle arrest, DNA repair, and apoptosis in response to DNA damage [5]. ATM is activated primarily by double-strand DNA breaks (DSBs). Upon DNA damage, it activates proteins like p53, Chk2, and BRCA1 to stop the cell cycle and induce the DNA repair mechanisms, or apoptosis if damage is too severe [8,9]. ATR is mainly activated by single-strand DNA (ssDNA) breaks, often at stalled replication forks, and plays a key role in responding to replication stress. It activates Chk1 and helps stabilize replication forks to maintain genome integrity. Both proteins are crucial for preventing mutations that could promote cancer. Inhibiting the ATM/Chk2 and ATR/Chk1 axes can sensitize tumor cells to DNA-damaging agents, particularly in tumors with DDR deficiencies [8,9,10]. Several small-molecule inhibitors targeting these proteins are in preclinical or clinical development and hold promise for the treatment of cancer patients in a more personalized and effective fashion than regular DNA-damaging chemotherapy agents. 

In this study we review the ATM/Chk2 and ATR/Chk1 axes in cancer and the new drugs developed to inhibit these proteins, which are being tested in different phases of preclinical experimentation or clinical trials. We also focus on recent discoveries that involve targeting these proteins in certain tumor genetic backgrounds related to DNA damage, whose inhibition leads to synthetic lethality. Finally, we address the role of ATM/Chk2 and ATR/Chk1 blockade as a way to enhance immunotherapy responses.

## 2. DNA Damage and Repair Mechanisms

### 2.1. Main Types of DNA Damage

Cells are continuously exposed to different agents that can damage integrity within DNA strands. Depending on the origin of these agents, DNA damage can be categorized as exogenous or endogenous [11]. Certain environmental hazards, toxic heavy metals, UV and ionizing radiation (IR), among others, have been described as common external agents responsible for DNA damage [12]. IR can directly induce a broad range of DNA alterations, such as single base lesions or single/double strand breaks (SSB, DSB) [13]. Additionally, one of the most harmful insult is due to an indirect effect of reactive oxygen species (ROS) caused by IR [13,14]. UV light is another source of exogenous DNA damage. UV light has been described to produce bulky lesions, such as cyclobutane-pyrimidine dimers (CPDs) and 6-4 photoproducts (6-4PPs), generate free radicals and induce DNA strand breaks [15,16].

On the other hand, cells may also suffer internal damage, which can generate multiple mutations or breaks, that can lead to base mismatches and genomic instability, among others. DNA polymerases are able to replicate 6 × 10^9^ nucleotides in every cell replication cycle [11,17]. Despite having high fidelity and proofreading activity, they can spontaneously make mistakes during the replication process. When cells are damaged during DNA replication (S-phase), replication forks temporally stop their activity (fork stalling). If maintained, this results in “fork collapse” leading to formation of DSBs and triggering DNA damage repair mechanisms [18]. Failure in DNA replication is mainly caused by DNA base damage. This includes abasic sites and alkylation processes [19,20]. Abasic sites (or apurinic/apyrimidinic, AP) can be spontaneously formed by destabilization of N-glycosyl bonds. If AP sites are maintained, they can cause blockade of transcription and DNA replication, leading to genomic instability. SSBs are easily formed, and if not repaired, DSBs lesions can appear [20]. Regarding DNA methylation, it is a reversible epigenetic process essential for maintaining DNA stability [18]. This process is catalyzed by a whole family of DNA Methyltransferases (DNMTs) [18], which transfer a methyl group from S-adenosyl-L-methionine (SAM) to the corresponding nitrogenous base. However, spontaneous methylation of DNA brings about the formation of cytotoxic adducts, such as 7-methylguanine (7-meG), 3-methyladenine (3-meA) and 3-methylguanine (3-meG) [19].

The diversity of DNA-lesions that may appear in every cell must be localized and repaired. Failures in recognition and repair systems leads to an increased number of mutations, which will increase the risk of cancer development [21].

### 2.2. Mechanisms of DNA Damage Response

#### 2.2.1. Single Strand Breaks

##### Mismatch Repair (MMR)

Errors in DNA replication typically include base-base mismatches and insertion-deletion loops. These errors are particularly frequent in repetitive DNA regions, such as microsatellites (Figure 1). Mismatches are repaired through the mismatch repair system (MMR), a highly conserved mechanism across species that involves several key proteins, including MLH1, MSH2, MSH6, and PMS2 in humans [22]. In prokaryotes, MutS gene detects mismatches in the double strand DNA by recruiting MutL, which allows downstream signaling with effector proteins. MutH belongs to type II family restriction endonucleases and is responsible for strand excision by creating a “nick”, from which the DNA helicase UvrD works to unwind DNA during recombination. There is a high similarity in the DNA repair mechanisms between prokaryotes and humans, with key roles of families MSH (MutS Homolog) and MLH/PMS (MutL Homolog/Post-Meiotic Segregation protein) proteins. The MSH complex plays a role in damage recognition, as MSH2-MSH3 and MSH2-MSH6 heterodimers slide on the DNA helix until mispaired bases are detected: single base mismatches or dinucleotide insertion-deletion distortions are recognized by the MSH2-MSH6 heterodimer, whereas larger insertion-deletion loops are detected by the MSH2-MSH3 heterodimer [23,24,25].

Upon MSH heterodimer binding to the DNA, there is a recruitment of proliferating cell nuclear antigen (PCNA), replication factor C (RFC) and MLH complex and exonuclease 1 (Exo1). The MLH complex is formed by MLH1-PMS2 heterodimer and is responsible for generation of a nick in the mutant, newly synthesized DNA strand, through its endonuclease activity [26,27,28]. Subsequently, Exo1 degrades the DNA between the error and the nick, DNA polymerase δ resynthesizes the strand and ligases seal the DNA sequence [26,27,29]. A failure in the MMR system can lead to microsatellite instability (MSI), characterized by widespread length variations in microsatellites (short, tandemly repeated DNA sequences scattered throughout the genome). MSI arises because insertion-deletion errors within microsatellites go uncorrected, leading to expansions or contractions of the repeat sequences [30]. 

##### Base Excision Repair (BER)

Genomic aberrations that do not distort the helix include depurination due to spontaneous hydrolysis and generation of oxidation derivatives from purines (8-oxoguanine and formamidopyrimines) and pyrimidines (thymine glycol and 5-OHU) [11,31,32] (Figure 1). The resulting modified bases are removed by DNA glycosylases through BER mechanisms, generating abasic (AP) sites that can be then replaced with the correct nucleotide [31,33,34]. One single damaged base (short patch repair, SP-BER) or a 2-8 nucleotide fragment (long patch repair, LP-BER) can be replaced [33].

This damage is principally detected by a DNA glycosylase that generates AP sites, which are then recognized by an AP endonuclease 1 (APE1), producing a SSB. This gap is engaged and protected by poly(ADP-ribose) polymerase 1 (PARP1). XRCC1 prevents excessive PARP1 binding. Then, DNA polymerase β along with ligase IIIα (for SP-BER) or DNA polymerases δ/ε along with PCNA/RFC (for LP-BER) cooperate to re-synthesize a corrected insert. The displaced damaged strand is removed by flap endonuclease 1 (FEN1) and the new segment is joint to the adjacent strands by DNA ligase I activity [35,36,37,38].

##### Nucleotide Excision Repair (NER)

Chemotherapeutic agents, environmental mutagens and UV irradiation induce bulky adducts that distort the structure of the double helix and that can be repaired by NER mechanisms (Figure 1). There are two subtypes within this repair system: the global genome NER (GG-NER), which repairs throughout the genome, and the transcription-coupled NER (TC-NER), a process that acts when the RNA polymerase in the process of transcription encounters this DNA damage and is halted. In this latter case, the repair system ensures an efficient transcription process [39]. Although different proteins are involved in each mechanism, both end up in the same pathway. In both GG-NER and TC-NER, the DNA lesion is recognized, the affected strand is excised and finally restored by polymerization/ligation using the non-damaged strand as a template [37].

In GG-NER, damage identification is carried out by multiple protein complexes such as UV-damaged DNA-binding protein (UV/DDB) and CETN2-XPC-RAD23B [40,41]. In contrast, in TC-NER, damage recognition occurs through CSA/CSB and RNA polymerase II, which stalls upon detecting the damage [42]. These sub-pathways then converge with the recruitment of transcription factor IIH (TFIIH), which interacts with XPB, to facilitate protein binding [43,44] and XPD, with helicase activity, to unwind DNA and form a bubble. XPA is necessary to verify that the damage has been correctly detected and acts as a scaffold protein [45,46]. Once the strands are separated, RPA binds to the single-stranded DNA to protect it from nuclease attacks during the repair process [47].

Finally, the endonucleases XPF and XPG remove the damaged fragment by performing 5’ and 3’ incisions, respectively [48], and DNA polymerases δ/ε resynthesize the missing DNA, which is ligated to the downstream DNA [49,50]. In this process, PCNA is responsible for the precise incorporation of nucleotides [49].

Although BER, NER, and MMR pathways involve excision steps that transiently generate SSBs as intermediates, these pathways are not primarily responsible for the repair of pre-existing SSBs. Instead, SSBs arising independently are typically resolved through direct ligation mechanisms or PARP1-dependent SSB repair pathways. Therefore, although BER, NER, and MMR have been included in the SSB-repair mechanisms, this classification should take into consideration this caveat.

## 3. Double Strand Breaks (DSB)

DSBs are directly or indirectly originated by environmental factors such as IR and certain drugs (Figure 2) [51]. These lesions are probably the most harmful among the different DNA injuries described earlier. Thus, organisms must have evolved mechanisms to detect and repair the genomic instability originated by DSBs [51]. There are two main mechanisms to repair DSBs: non-homologous end joining (NHEJ) and homologous recombination (HR). 

### 3.1. Non- Homologous End Joining (NHEJ)

Classical non-homologous end joining is initialized with dimerization of Ku70-Ku80 within the DSB site [52] (Figure 2). This binding will provide a scaffold for recruitment of DNA-PKs, DNA ligase IV, XRCC4, XRCC4-like factor (XLF) and PAXX (an XRCC4-like factor that will stabilize the NHEJ complex on the damaged chromatin) [53]. There is evidence suggesting that the kinase activity of DNA-PKs, together with the XRCC4-XLF complex are crucial in maintaining the two broken ends of DNA close enough to ligate both strands [53,54]. DNA-PK phosphorylates and activates Artemis, which has 5’ nuclease activity, needed to eliminate DNA strand overhangs, thus generating blunt ends [55]. For gap filling, DNA polδ and DNA polµ (members of Pol X family) are required [55] and finally, ligation of both strands is completed by the XRCC4-Ligase IV complex. When the ligation process is ended, the NHEJ complex is dissolved.

Apart from the classical NHEJ process explained, there is an alternative pathway for NHEJ. This pathway operates when the classical pathway is compromised or unavailable [56,57]. It typically uses short regions of microhomology (2–25 base pairs) flanking the break to align the DNA ends before joining and is a slower and more error-prone repair mechanism compared to classical NHEJ. The alternative pathway involves enzymes that cause end resection, nucleases capable of removing non-compatible 5’ and 3’ overhangs and ligases. This mechanism of DNA repair is less known, but it can be explained as follows: PARP-1 is used as a sensor of the DSBs and catalyzes the activation and recruitment of repair proteins. Enzymes MRE11, CtIP, and Exo1 initiate a limited 5′ to 3′ resection. The exposed single-stranded regions search for and align at short regions of microhomology, which helps stabilizing the ends for ligation. The aligned ends are annealed and the non-matching sequences are removed by endonuclease FEN1. Finally, DNA polymerase Pol θ fills in the gaps, in cooperation with the ligase III-XRCC1 complex.

### 3.2. Homologous Recombination Repair (HRR)

Homologous recombination represents the second pathway of DSB repair. Unlike NHEJ, in HR a sister chromatid is needed, thus providing a high fidelity, error-free process [52,58] (Figure 2). As HR pathway is limited by the presence of a sister chromatid, this mechanism of repair is constrained to the S/G2 phase of the cell cycle [52].

DNA damage is sensed by the MRN complex (comprising MRE11, RAD50 and NBS1) and 53BP1 (sensor of changes in chromatin structure), which consequently activate ATM (Figure 2). ATM phosphorylates γH2AX and activates downstream pathways leading to cell cycle arrest via Chk2, while the repair process takes place. γH2AX, MDC1 and BRCA1 participate in the sensor/effector complexes necessary for the repair mechanism. Then, the DNA is resected, leaving 3’ overhangs. This process is necessary to invade in the homologous DNA strand. The resulting ssDNA, upon resection, is coated with RPA complexes, which in turn activate ATR, leading to Chk1-mediated cell cycle arrest. RPA is eventually replaced by Rad51 and DMC1 through BRCA2, which triggers the initiation of homologous recombination [52,59]. The invasion strand serves as template for DNA synthesis, where DNA polymerase δ and PCNA play a key role [59]. Finally, after the homologous recombination process, the heteroduplex is dissociated.

## 4. The ATR/Chk1 and ATM/Chk2 Axes in DNA Damage Repair

Participation of ATR/Chk1 and ATM/Chk2 kinases is crucial in DDR, as regulators of the repair machinery. Downstream signaling of these axes results in either halt of the cell cycle, which allows for DNA repair, or apoptosis and subsequent cell death when the damage is deemed irreparable [8].

### 4.1. ATR/Chk1 Signaling

The ATR/Chk1 pathway is key in ensuring the integrity of DNA replication; as such, it is triggered by replication stress and a wide range of genotoxic events. In fulfilling its protective function, ATR/Chk1 signaling often culminates with halt of the cell cycle during the replicative or S phase [60]. ATR is a serine/threonine kinase that phosphorylates a wide range of effector molecules in response to replication hindrances or DNA damage. ATR binds to RPA-coated ssDNA overhangs through interaction with ATRIP (ATR-interacting protein) [61]. This complex is characteristic of stressed replication forks, as well as certain intermediaries in DNA repair mechanisms, such as HR or NER [8]. ATR binding promotes stabilization of the stalled replication fork in order to prevent imminent collapse. Moreover, upon allosteric stimulation by TOPBP1 (Topoisomerase Binding Protein 1) or ETAA1 (Ewing’s tumor-associated antigen 1), ATR can phosphorylate and modulate key substrates involved in cell cycle regulation and HR [60,62] (Figure 3).

Chk1 is the main downstream effector of ATR. Activated upon its phosphorylation by ATR, Chk1 inhibits Cdc25A and C phosphatases to arrest the cell cycle at intra-S and G2/M checkpoints, respectively [62]. Cdc25C is then ubiquitinated and degraded by the proteasome, followed by decreased CDK (Cyclin-Dependent Kinase) activity, which delays cell cycle progression until the damage has been repaired [53]. In parallel, by means of Wee1 activation, Chk1 maintains CDK and Cyclin suppression and reinforces cell cycle arrest [63].

Beyond the described mechanisms of DNA repair and cell cycle regulation, ATR/Chk1-mediated phosphorylation of p53 and other transcription factors actively contributes to the increased expression of genes related to DNA repair and survival [64]. The ATR/Chk1/Wee1 cascade also contributes to HR by interaction with ssDNA overhangs generated from end resection in the intermediate steps of repair. In this context, ATR has been proven to recruit Rad51, while Chk1 can phosphorylate both this key substrate and BRCA2. Additionally, Wee1-induced CDK suppression promotes this mechanism of DSBs repair [62]. Other unexpected roles of this axis include preventive suppression of chromosome instability and faithful segregation due to mitosis-specific centromeric R-loop formation [65]. Lastly, ATR/Chk1 axis supports and fires normal replication fork origins in unperturbed S phase, ensuring reliable replication and preventing further damage altogether [66,67].

### 4.2. ATM/Chk2 Signaling

The ATM/Chk2 pathway primarily responds to DSBs in the DNA, which can be induced by IR, oxidative stress and certain forms of replication errors. In this response, ATM is recruited by the MRN complex at the site of DSB, triggering a regulation cascade that initializes with its autophosphorylation [53,68,69]. ATM’s kinase activity is key in the activation of Chk2, p53 and several other downstream targets that orchestrate cell cycle progression [53,70]. Moreover, ATM directly contributes to NHEJ and HR pathways by recruitment of repair proteins, such as DNA-PKcs or BRCA1 to the damage site [68]. In immediate response to DSBs, the Ser139 residue in the C-terminal tail of histone H2AX is rapidly phosphorylated by ATM, serving as a docking site for chromatin remodeling and DNA repair complexes [53]. Importantly, the consequently recruited factors will determine the repair mechanism to get activated in response to the detected DSB. The competition between 53BP1/RIF1 and BRCA1/CtIP is crucial in this decision-making process for both DNA damage and replicative stress. While, 53BP1 and its cofactors mediate DNA end protection enabling repair by NHEJ, its antagonist BRCA1 promotes end resection and allows HR repair by recruiting the endonuclease CtIP [71,72]. This regulation appears to be cell cycle dependent, with 53BP1-mediated NHEJ taking place at all stages and CtIP-initiated HR being conditioned to the S/G2 phase due to its dependence on a homologous region to ensure faithful DNA repair [68,71]. Enforcing the established convergence of DDR pathways, 53BP1 is a cell cycle regulator itself, responding to mitotic stress and DNA damage by stabilizing p53 [73]. 

ATM signaling initiates the p53 checkpoint pathway, leading to cell cycle arrest at the G1/S transition, ultimately triggering senescence or apoptosis in cases of severe and irreparable damage [70]. Additionally, ATM activation can also regulate the G2/M checkpoint, ensuring DNA stability for successful mitosis. Downstream of ATM, Chk2 contributes to p53 signaling by phosphorylating specific residues and thus maintaining cellular arrest through stabilization and activation of this tumor suppressor [70]. The induction of the p53 transcriptional target p21 halts the cell cycle to allow DNA repair without further progress through the cycle [63,70,74]. In addition, the cofactor Strap (stress responsive activator of p300), needed for an effective p53 response, is also a target of ATM and Chk2. Phosphorylation of distinct residues in Strap by this axis contributes to its stabilization and nuclear accumulation, which, in turn, enhances p53 half-life and signaling [75].Concurrently, Cdc25A and C kinases are inhibited by Chk2, which results in further cell cycle arrest by depletion and absence of CDKs and cyclin stimulation [69,76]. Interestingly, Chk2-dependent phosphorylation of Cdc25A on Ser 123 upon irradiation has been described to not only contribute to IR-induced DNA damage repair by S-phase delay, but also as a preventive mechanism against accumulation of mutations that confer radioresistance [77]. 

Concurrently, Cdc25A and C kinases are inhibited by Chk2, which results in further cell cycle arrest by depletion and absence of CDKs and cyclin stimulation [69,76]. Interestingly, Chk2-dependent phosphorylation of Cdc25A on Ser 123 upon irradiation has been described to not only contribute to IR-induced DNA damage repair by S-phase delay, but also as a preventive mechanism against accumulation of mutations that confer radio-resistance [77].

Aside from intervening in cell cycle arrest, the ATM/Chk2 axis prevents chromatin breakage via the phosphorylation of the scaffold protein INCENP, which promotes its binding to the chromosomal passenger complex (CPC) in cytokinesis, establishing an abscission checkpoint and avoiding DNA damage in the process [78]. In addition to the tumor suppressive character of ATM/Chk2 signaling, recently, CCNDBP1 (cyclin D1 binding protein 1), a protein associated to chemotherapy-induced damage rescuing, has been found as a possible linker between the ATM/Chk2 signaling and chemoresistance [79].

### 4.3. Cross-Talk Between ATR/Chk1 and ATM/Chk2

The extensive study of Chk1/ATR and Chk2/ATM axes has uncovered the complex network of overlapping and non-redundant interactions that conform DDR (Figure 3). Aside from converging in the established downstream inhibition of Cdc25 phosphatases, the main components of these pathways can exert direct modifications and regulate mutually [53]. This interaction was reported both ways, upon UV-induced DDR in ATR-dependent ATM activation and, vice versa, in IR-induced ATM-induced activation of ATR [80,81]. Further cross-talk reveals connecting points towards the p53 downstream signaling pathway. As an example, the indispensable role of Chk1 in maintaining ATM/p53/p21-triggered G2 cell cycle arrest has been observed in the context of sustained damage [64]. 

For all the reasons mentioned, the pharmaceutical industry identified the ATR/Chk1 and ATM/Chk2 kinases as highly convenient targets to develop anticancer drugs, particularly in the context of certain mutations related to DDR. Many cancers exhibit defects in DDR, making them more reliant on ATR/Chk1 or ATM/Chk2 for survival. Inhibition of these pathways may selectively kill cancer cells by enhancing genomic instability and preventing repair of DNA damage induced by chemotherapy or radiation. ATR/Chk1 inhibition can be particularly effective in tumors with replication stress, whereas ATM/Chk2 inhibition may sensitize tumors with pre-existing DDR deficiencies, making them more susceptible to DNA-damaging agents. Also, dual inhibition or combination with other therapies, such as PARP inhibitors, can result in synthetic lethality. Moreover, increased DNA damage can lead to generation of neoantigens and enhanced response to immunotherapy. In the next section we describe the main ATR, ATM, Chk1 and Chk2 inhibitors used in preclinical experiments and/or clinical trials and explain how these drugs, alone or in combination, may help in cancer treatment.

### 4.4. ATR Inhibitors

As previously mentioned, ATR is deeply involved in responses against single-strand DNA damage or replication stress. Therefore, cancer cells that face genomic instability will more likely depend on the ATR pathway for survival [82]. Among the drugs undergoing clinical testing, berzosertib (M6620, VX-970) has been studied as a single agent or in combination with chemotherapeutic agents (gemcitabine, topotecan, carboplatin, cisplatin) [83]. In preclinical studies of patient-derived lung tumor xenografts, combination of berzosertib with cisplatin led to a substantial tumor regression and delayed regrowth [83]. These results guided different phase I clinical trials, such as the CHARIOT trial (NCT03641547), where berzosertib was evaluated in combination with radiotherapy for palliative treatment of oesophageal cancer, or with cisplatin-capecitabine chemotherapy in solid tumors. Berzosertib was well tolerated and encouraging clinical activity was observed, as 5 patients presented partial responses and 10 showed stable disease [84]. Another phase I trial (NCT02487095) assessed the efficacy of berzosertib in combination with topotecan in the second-line treatment of small cell lung cancer (SCLC) patients [85]. The trial demonstrated tolerability of berzosertib combined with topotecan and showed promising clinical activity in refractory SCLC, with 2 confirmed partial responses and 8 cases of prolonged stable disease [85]. In a phase II study where berzosertib was evaluated in combination with gemcitabine, in platinum-resistant ovarian cancer (NCT02595892), increased median progression-free survival (mPFS) was observed in the gemcitabine plus berzosertib group (22.9 weeks) compared with the gemcitabine group (14.7 weeks) [86].

AZD6738 (ceralasertib) is anotherATR inhibitor that has shown antitumor activity in combination with DNA-damaging anticancer agents [8], such as cisplatin, causing rapid regression of ATM-deficient non-small cell lung cancer (NSCLC) xenograft models [87]. A phase I clinical trial studying the effect of combining ceralasertib and paclitaxel (NCT02630199) determined that the treatment was well tolerated and demonstrated promising antitumor activity, especially in patients with advanced melanoma resistant to anti-PD1/L1 treatments [88]. A Recommended Phase 2 Dose in this trial determined an optimal dose of 240 mg twice daily on days 1–14, combined with paclitaxel at 80 mg/m² on days 1, 8, and 15 of each 28-day cycle [88]. Among the 57 patients treated, the overall response rate (ORR) was 22.6%, with a higher ORR (33.3%) for the 33 patients with melanoma resistant to prior anti-PD1/L1 therapy (mPFS 3.6 months, mOS 7.4 months) [88]. In the CAPRI phase I trial (NCT03462342), patients received olaparib 300 mg twice daily and ceralasertib 160 mg on days 1 to 7 of a 28-day cycle [89]. Six patients presented partial responses, yielding an ORR of 50%. Combination of olaparib and ceralasertib was found to be tolerable and demonstrated activity in HR-deficient, platinum-sensitive recurrent high-grade serous ovarian cancer (HGSOC) patients who had progressed after benefiting from PARP1 inhibition as their penultimate treatment [89]. 

BAY1895344 (elimusertib) is an oral ATR inhibitor that has demonstrated antitumor efficacy as monotherapy in CDX models and has shown synergistic effects when combined with the PARP1 inhibitor olaparib *in vivo* [90]. In a phase I dose escalation trial (NCT03188965) where elimusertib was administered at doses ranging from 5 to 80 mg, twice daily, to 21 patients with advanced solid tumors, the maximum tolerated dose was determined to be 40 mg twice daily with a 3-day-on/4-day-off schedule. 4 patients presented partial responses and stable disease was noted in 8 patients, with a median duration of response of 315.5 days. In fact, patients that responded had ATM protein loss and/or deleterious mutations of *ATM* [91]. Currently, other trials studying the combination of elimusertib with other agents (pembrolizumab, NCT04095273) are being conducted [91]. In Table 1 we show a selected representation of ATR inhibitors and their evaluation in preclinical and/or clinical trials.

In spite of the current promising landscape, several studies point at the potential translational issues of ATR/Chk1 inhibition in tumors that are not subject to sufficient replication stress. This ultimately highlights the possibility of resistance to treatment and the need for downstream effector suppression [92]. In this context, high-replication stress models, such as *KRAS*-mutated cancers, should be considered as relevant targets for ATR/Chk1 inhibition. Additionally, SMG8/9-defective gastric cancer can evade the antitumor effects of ATR inhibitors by causing a reduction in the induced transcription and replication conflicts [93]. Another biomarker of ATR inhibition resistance is Cdc25A. As a downstream effector for the ATR/Chk1 axis, Cdc25A overexpression bypasses the effects of ATR inhibition, notably minimizing sensitivity to treatment [94].

### 4.5. ATM Inhibitors

When DSBs appear, ATM is activated by phosphorylation to repair DNA breaks [95]. As a master regulator in DNA damage repair, ATM has been widely studied as a therapeutic target [8] (Table 1). KU-55933, the first ATM inhibitor developed, suppressed downstream phosphorylation events triggered by ionizing radiation. The inhibition increased cell sensitivity to radiation and DNA-damaging chemotherapy drugs (etoposide, doxorubicin and camptothecin), and disrupted radiation-induced cell cycle arrest [96]. KU-60019 is an optimized analogue of KU-55933, developed to overcome limitations dealing with high lipophilicity and poor bioavailability [8,97]. With improved aqueous solubility and pharmacokinetics, KU-60019 is a more effective and specific ATM kinase inhibitor [8]. It can radiosensitize human glioma cells at low micromolar concentrations and, notably, also appears to reduce glioma cell motility and invasion even in the absence of radiation [98]. Despite its promising *in vitro* activity, the low bioavailability of this class of compounds remains a barrier to clinical application [8].

Preclinical studies of ATM inhibition with AZD0156, which presents high oral bioavailability, have demonstrated that it improves olaparib’s efficacy in two patient-derived triple-negative breast cancer xenograft models [8]. AZD0156 alone was ineffective to inhibit cancer cell growth in colorectal cancer PDX models, but increased antitumor effects were observed when combined with irinotecan [99]. However, a phase I clinical trial (NCT02588105) combining AZD0156 with olaparib or irinotecan did not meet the expected results [99].

M3541 is a recently developed oral ATM inhibitor that, in preclinical studies, enhanced the sensitivity of various tumor cell lines to ionizing radiation and a topoisomerase inhibitor [8]. In mouse models with human tumor xenografts, M3541 combined with radiation led to complete tumor regression by inhibiting ATM activity and its downstream signaling [100]. Despite promising preclinical results, a Phase I clinical trial combining M3541 with palliative radiotherapy was discontinued early due to a lack of dose-response and poor pharmacokinetics [101]. As a result, further clinical development of M3541 was not pursued [101].

AZD1390 is another ATM-targeting DDR inhibitor optimized to present blood-brain barrier permeability [102]. In preclinical models, including syngeneic and glioma PDX, as well as orthotopic lung-brain metastatic models, AZD1390 administered with daily doses of IR significantly resulted in tumor regressions and improved animal survival compared to radiation treatment alone [102]. Currently, there are several Phase I clinical trials actively recruiting participants, such as: the CONCORDE trial (NCT04550104) in patients with NSCLC in combination with conventional radiotherapy [103]; and NCT03423628, to assess the safety and tolerability of AZD1390 in combination with radiation therapies in patients with glioblastoma multiforme and patients with brain metastases [104].

### 4.6. Chk1/2 Inhibitors

Inhibiting Chk1 and Chk2, highly conserved serine/threonine kinases and key downstream targets of ATR and ATM, respectively, could disrupt the initiation of G1/S and G2/M checkpoints, impair DNA repair mechanisms and promote apoptosis in tumor cells [9,105]. LY-2606368 (prexasertib) has shown promising results in preclinical models. For instance, it was described that Chk1/2 inhibition in monotherapy or in combination with cisplatin was able to significantly reduce tumor growth in a syngeneic model of SCLC [106]. Moreover, it increased cisplatin and olaparib potential and improved the response in platinum-resistant models [106]. A Phase I trial using prexasertib combined with olaparib showed preliminary clinical activity in *BRCA*-mutant patients with HGSOC, who had previously progressed on a PARP inhibitor. Notably, 4 of 18 patients with *BRCA1*-mutant PARP inhibitor–resistant HGSOC, achieved partial responses, suggesting potential for this combination therapy in overcoming resistance [107]. Phase II clinical trials with prexasertib have also shown promising results, particularly in ovarian cancer. In the NCT02203513 trial, prexasertib was particularly effective in patients with platinum-resistant or refractory *BRCA*-wild type HGSOC, with an 8/24 partial response rate [108]. The NCT03414047 trial reported an ORR of 12.1% in platinum-resistant patients and 6.9% in platinum-refractory patients, with durable activity as a single agent [109]. Phase II studies have also been conducted in other diseases, including triple-negative breast cancer and SCLC, but the efficacy observed in these cases was lower compared to that of ovarian cancer [110,111].

In a preclinical study evaluating the effects of Chk1 inhibition by GDC-0575, *in vivo* experiments were conducted using soft tissue sarcoma PDX models. The combination of GDC-0575 with gemcitabine demonstrated a synergistic antitumor effect and improved survival in dedifferentiated liposarcoma and leiomyosarcoma xenografts and *TP53* mutant PDX models with the combination therapy, compared to either agent alone (no reduction was observed in the *TP53* wild type PDX model) [112]. A phase I clinical trial using this drug in combination with gemcitabine (NCT02797964) demonstrated modest tolerability but improved antitumor activity. 4 patients achieved confirmed partial responses, 4 presenting *TP53*-mutated tumors. Overall, 15% of patients had stable disease for a period over 4 months. While GDC-0575 was in general safe administered both as a monotherapy and in combination with gemcitabine, hematological toxicities were frequent yet manageable [113]. 

The preclinical evaluation of PHI-101, a novel Chk2 inhibitor, demonstrated its potent anticancer activity against refractory ovarian and breast cancer cells, with improved efficacy both as monotherapy and in combination, *in vitro* and *in vivo* [114]. Currently, a phase Ia trial named CREATIVE (NCT04678102), is being conducted with the objective of assessing the dose-limiting toxicity (DLT) and the maximum tolerated dose (MTD) of PHI-101 in recurrent epithelial peritoneal, fallopian or ovarian cancer [115]. 

*In vitro* studies across various human tumor cell lines have demonstrated the effectiveness of Chk2 inhibition with CCT241533 in response to DNA damage [116]. Notably, CCT241533 potentiated the cytotoxic effects of PARPi, such as olaparib, in cell viability assays, but in contrast, it did not significantly potentiate the anticancer effects of other genotoxic agents like etoposide, bleomycin or gemcitabine [116]. As a result, combining the Chk2 CCT241533 inhibitor with PARPi may offer a novel strategy for targeted cancer treatment [116]. Table 1 summarizes relevant results with Chk1/2 inhibitors.

## 5. Synthetic Lethality Approaches to Target the ATR/Chk1 and ATM/Chk2 Axes

As described before, synthetic lethality implies a therapeutic approach by which the simultaneous alteration of two or more genes, or targeting two or more proteins in complementary pathways related to cell survival, results in vast cell death, while alteration of either gene/protein alone has little or modest effect on viability [117]. This type of perturbations are being widely studied along with DNA repair deficiencies in different cancers, receiving the name of “BRCAness”, as it was first described with the discovery of *BRCA* gene mutations in breast tumors, which were highly sensitive to PARP inhibitors [117,118]. In this section, we summarize some of the most recent synthetic lethality applications to the ATR/Chk1 and ATM/Chk2 regulatory pathways. 

Weaknesses in cancer cells that present DNA repair deficiencies or genomic instability can be exploited for synthetic lethality with Chk1/ATR and Chk2/ATM inhibitors [119,120]. ATR/Chk1 signaling, hyperactive in cancer cells as a result from replication stress and uncontrolled proliferation, offers a good opportunity for synthetic lethality [62]. This could be particularly relevant in *BRCA*-deficient tumors [121,122]. Among the various preclinical and clinical trials conducted so far, ATR/Chk1/Wee1 inhibition has been described to be synergistic with the PARP inhibitor rucaparib [122]. Some studies even showcase successful overcoming of resistance to the PARP inhibitor olaparib in *BRCA2*-mutant ovarian cancer, by targeted inhibition of ATR/Chk1 [123]. Similarly, *TP53* deficiency-induced loss of G1 checkpoint, makes cancer cells particularly vulnerable to ATR/Chk1 inhibition. Sensitivity to elimusertib and other ATR inhibitors is being thoroughly studied, with these drugs having readily entered clinical trials for *TP53*-deficient triple-negative breast cancer treatment, following on promising preclinical data [124,125].

ATR/Chk1 inhibition has also been used in combination with chemo- and radiotherapy. Potential synergy in cancer cell cytotoxicity has been studied by combining ATR inhibitors AZD6738 or M6620 with a wide arrange of chemotherapeutic agents, from cisplatin and carboplatin to topotecan and gemcitabine. These strategies have shown promising results in preclinical assays, but high toxicities in clinical trials [126]. Interestingly, simultaneous ATR and Chk1 inhibition was synthetic lethal in preclinical studies involving a variety of solid tumor models, with improved effects with hydroxyurea treatment [121]. ATR blockade induced synthetic lethality and overcame chemoresistance in *TP53*- or *ATM*-defective chronic lymphocytic leukemia cells [127]. Using the ATR inhibitor AZD6738, authors showed that cells accumulated unrepaired DNA damage, ultimately leading to cell death by mitotic catastrophe. Inhibition of ATR induced synthetic lethality in MMR-deficient cells and increased the effect of immunotherapy, in the CT26 colorectal cancer model [128]. By using a model of *ATR*-deficient DLD-1 human colorectal cancer cells, Schneider et al., [129] demonstrated synthetic lethality upon POLA1 inhibition. siRNA-mediated *POLA1* depletion sensitized several cancer cell lines to ATR and Chk1 inhibitors. ATR inhibition was also shown to be synthetic lethal with *ERCC1* deficiency [130]. A synthetic lethal screening discovered increased sensitivity to ATR inhibitors in mantle cell lymphoma with *ATM* loss-of-function [131]. In an orthotopic breast cancer model, co-targeting ATR and Wee1 led to tumor-selective synthetic lethality, with tumor remission and metastasis impairment. Mechanistically, this combination left cells with unrepaired or under-replicated DNA, thus inducing mitotic catastrophe [132].

Synthetic lethality upon combination of Chk1 (SRA737) and Wee1 (AZD1775) inhibitors has also been reported in castration-resistant prostate cancer [133]. In *EZH2* deficient T cell acute lymphoblastic leukemia (T-ALL), a synthetic lethal screening identified Chk1 inhibition as an exploitable vulnerability. *EZH2* loss was related to a transcriptomic signature in immature T-ALL cells, characterized by upregulation of *MYCN* and replication stress [134]. ATR-, Chk1- and Wee1-targeting drugs caused HR repair deficiency and induced synthetic lethality with PARP inhibitors [122]. Combination of an HDAC8 inhibitor with the Chk1-targeting drug AZD-7762 produced synthetic lethality in preclinical cancer models. HDAC8 impairment slowed DNA fork progression and induced Chk1/2 activation, allowing the co-targeting strategy that resulted in cell death [135]. A scheme summarizing some key synthetic lethal interactions involving ATM, ATR, Chk1 and Chk2 is shown in Figure 4.

Loss of *BRCA1* has also been shown to be synthetic lethal with *ATM* loss [136], suggesting that drugs targeting ATM can be highly effective against tumors in the context of *BRCA1* mutation. Synthetic lethality between epigenetic silencing of *BEND4*, a DNA repair gene, and ATM inhibitors has been described in pancreatic ductal adenocarcinoma (PDAC). Promoter methylation of *BEND4* was found in 58.1% PDAC patients. BEND4 is involved in NHEJ signaling and loss of *BEND4* significantly increased the sensitivity of PDAC cells to the ATM inhibitor AZD0156 [137]. Oh et al., [138] described a synthetic lethal strategy using PARP and ATM inhibitors to overcome trastuzumab resistance in HER2-positive cancers. The authors described increased PARP1 levels in trastuzumab-resistant cells. Inhibition of PARP1 with olaparib in the resistant cells restrained proliferation, but activated ATM to maintain genome stability. Dual inhibition of PARP and ATM (with AZD0156) in this context caused synthetic lethality. Ratz et al., [139] showed that combination between an EZH2 inhibitor (GSK126) and an ATM inhibitor (AZD1390) is synthetic lethal in *BRCA1*-deficient breast cancer. Lethality was manifested in reduced colony formation, increased genotoxic stress, and apoptosis-mediated cell death *in vitro*, as well as significantly increased anti-tumor activity *in vivo*. 

These and other published results exemplify the opportunity to discover novel cancer vulnerabilities by pharmacological combinations targeting ATM, ATR and Chk1/2, especially in DDR impaired genetic backgrounds. 

## 6. Synergistic Anticancer Effects Elicited by Combining ATM/Chk2 or ATR/Chk1 Inhibition with Immunotherapy

DDR inhibitors have been shown to enhance antitumor immune responses and potentiate immunotherapy. The rationale for this combination is that DDR inhibitors can increase tumor immunogenicity by promoting accumulation of DNA damage, leading to the presence of cytosolic DNA fragments that activate the cGAS-STING pathway. This activation triggers type I interferon responses and upregulation of inflammatory cytokines and chemokines, enhancing recruitment and activation of immune cells, particularly cytotoxic T cells [140]. Examples of this cooperation are copious and summarized below.

The recent link between ATR/Chk1 and modulation of the tumor microenvironment (TME) has sparked interest in the combination of ATR inhibitors with anti-PD-(L)1 therapy [141]. Several ongoing early-phase clinical trials in solid tumors are studying the administration of ATR-targeting drugs (M6620, AZD6738) together with either avelumab or pembrolizumab [142]. The ATR inhibitor ceralasertib potentiated checkpoint-based immunotherapy by up-regulation of type I interferon (IFNI) pathways [143]. Similar findings were reported by Taniguchi et al., in SCLC, using the ATR inhibitor berzosertib [144]. In hepatocellular carcinoma, addition of the ATR targeting drug AZD6738 to radioimmunotherapy boosted the immune cell infiltration and enhanced interferon (IFN)-γ production [145]. ATR inhibition also activated STING and increased tumor expression of MHC-I [146]. Radiation therapy in combination with the ATM-targeting drug AZD0156 increased STING-dependent anticancer responses. A low-dose of this drug plus radiotherapy synergistically increased IFN-β, MHC-I and PD-L1 expression in tumor cells [147]. In lung cancer, combination of the ATR inhibitor berzosertib with ablative radiotherapy remodeled the TME and enhanced immunotherapy responses [148]. In this study, authors showed that ATR inhibition increased radiation-induced damage, with activation of the cGAS/STING pathway. Adding immunotherapy further enhanced antitumor and antimetastatic effects. In a phase I clinical trial, the Chk1 inhibitor prexasertib combined to anti-PD-L1 therapy showed evidence of CD8 + T-cell activation in peripheral blood in response to treatment [149]. In preclinical cancer models, combination of the oral Chk1 Inhibitor SRA737 with low-dose gemcitabine enhanced the effect of anti-PD-L1 therapy by modulating the TME. This treatment led to increase in CD8+ cytotoxic T cells, dendritic cells, and M1-like macrophages and IFNβ, CCL5 and CXCL10, as well as decrease in M2-like macrophages and MDSC [150]. 

Inhibition of the ATM/Chk2 axis also induced cGAS/STING signaling in *ARID1A*-deficient tumors [151]. ARID1A is a member of the chromatin-remodeling complex SWI/SNF, which is frequently mutated in cancer. Using data from the Cancer Genomic Atlas, authors found augmented expression of Chk2 in *ARID1A*-mutated/deficient tumors. Inhibition of ATM/Chk2 led to replication stress, accumulation of cytosolic DNA and activation of the STING-mediated innate immune response. This resulted in increased tumor-infiltrating lymphocytes [151]. 

These examples show the mechanistic and clinical rationale for combining ATR/Chk1 and ATM/Chk2 to enhance the effect of immunotherapy. Clinical trials co-targeting these proteins and other DDR proteins are underway, with great expectation about the potential synergism in anticancer therapy [152].

## 7. Conclusions and Future Perspectives

Drugs that cause DNA damage are still pillars for cancer treatment. Although DNA damage induced by anticancer drugs can be repaired by different mechanisms to maintain genomic integrity, dysfunctionality of these pathways in cancer cells increases susceptibility to DDR-based treatments. Central to DDR are kinases ATM, ATR, Chk1, and Chk2, which orchestrate DNA repair, cell cycle arrest, or apoptosis upon DNA damage. Inhibiting these kinases can sensitize tumor cells to treatments, including chemotherapy, radiotherapy, and DNA repair-targeted therapy, especially in DDR-deficient tumors. Genomic studies and pharmacological inhibitors targeting ATM/Chk2 and ATR/Chk1 axes have identified novel synthetic lethality approaches, leading to a pronounced antitumor growth. These strategies are revealing novel options for personalized medicine, in some cases being currently tested in clinical trials. Furthermore, ATR/Chk2 and ATM/Chk1 inhibitors enhance immunotherapy by increasing tumor immunogenicity through activation of immune pathways, such as cGAS-STING, increase in mutation load and promotion of T-cell infiltration, particularly when combined with immune checkpoint inhibitors. Therefore, inhibition of ATM, ATR, Chk1, and Chk2 represents a promising strategy in modern oncology. By exploiting these cancer-specific weaknesses, the therapeutic approach based on ATR/Chk2 and ATM/Chk1 inhibition is expected to improve efficacy, overcome resistance, and reduce toxicity, potentially increasing patients’ survival.

However, these therapies also face several challenges. One major issue is toxicity, as these kinases play vital roles in normal cell DDR, potentially leading to adverse effects in healthy tissues. In fact, relevant toxicity has been reported in several clinical trials. Selectivity is another hurdle to maximize therapeutic effects. Identifying patients whose tumors have specific DDR deficiencies seems to be highly important for therapeutic success, something that many times is not considered in clinical trials. Biomarker development remains limited, complicating patient stratification. Resistance mechanisms, including pathway redundancy and mutations, can quickly emerge and tumor heterogeneity further complicates response prediction. Drug combinations may enhance efficacy against resistance mechanisms, but they can also exacerbate toxicity. Overall, precision targeting of DDR pathways requires careful balancing of efficacy and safety.

## Figures and Tables

**Figure 1 cells-14-00748-f001:**
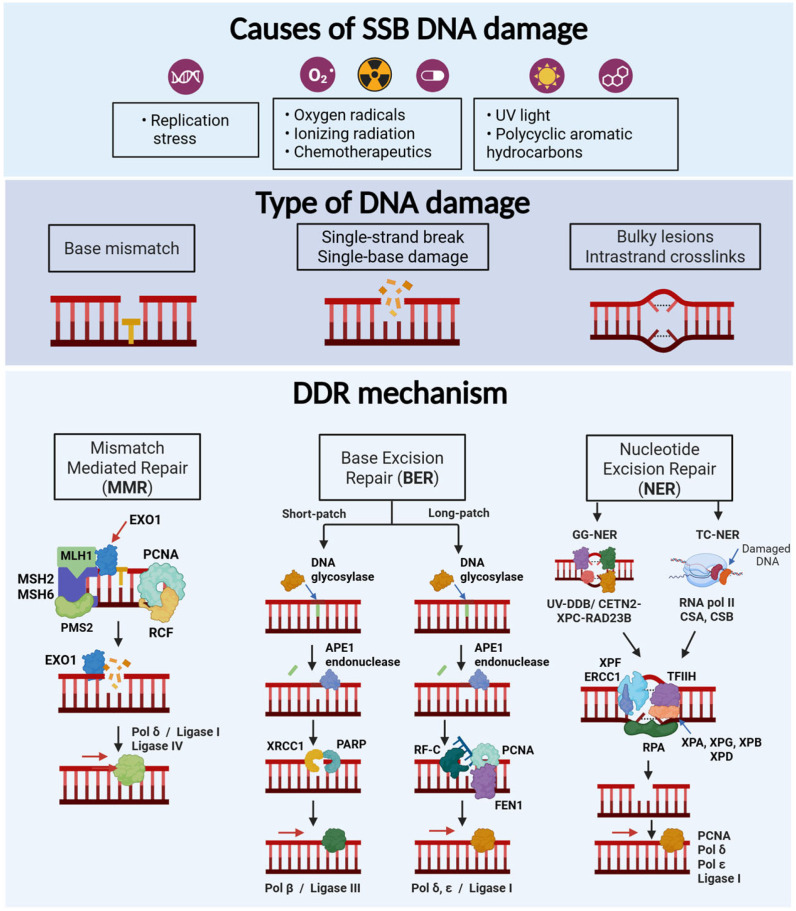
Scheme showing the different causes and types of DNA damage induced by single-strand breaks (SSBs), together with their corresponding DNA damage response mechanisms. Agents causing SSBs include base mismatch, base damage or bulky lesions, which are repaired by mismatch mediated repair (MMR), base excision repair (BER) or nucleotide excision repair (NER), respectively.

**Figure 2 cells-14-00748-f002:**
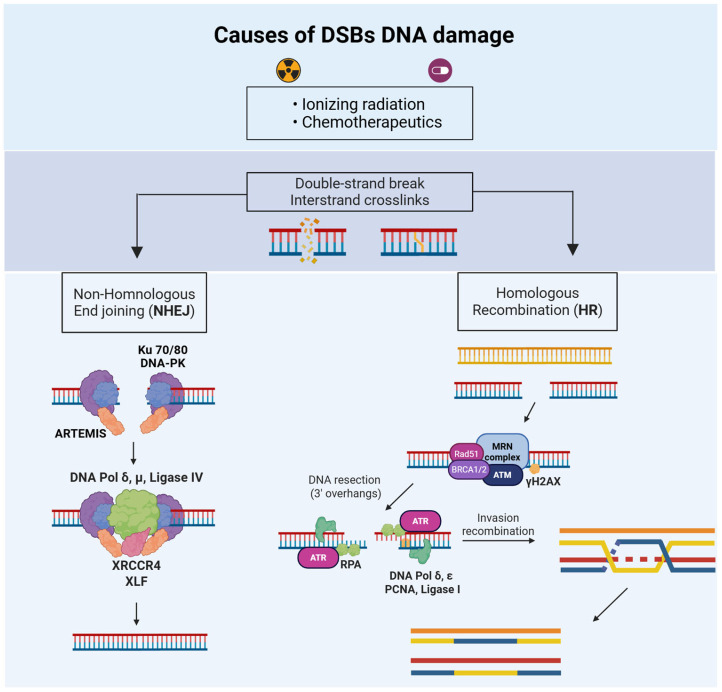
Scheme showing causes and types of DNA damage induced by double-strand breaks (DSBs), together with their corresponding DNA damage response mechanisms. DBBs can be caused by ionizing radiation or certain drugs. The repair process can be performed by the non-homologous end joining (NHEJ) process or by homologous recombination (HR). Only a handful of relevant proteins are represented in the scheme, albeit many more ones participate in these repair mechanisms, especially in HR.

**Figure 3 cells-14-00748-f003:**
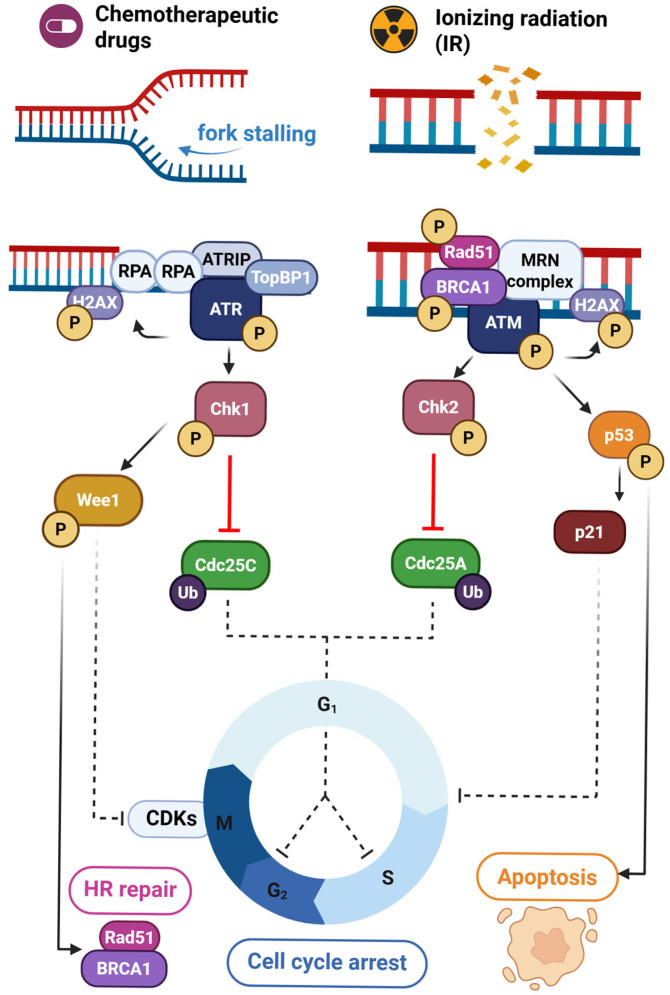
Crosstalk between Chk1/ATR and Chk2/ATM. Replication stress and ionizing radiation can induce activation of Chk1/ATR and Chk2/ATM pathways, respectively, converging in cell cycle arrest. In addition, the Chk1/ATR/Wee1 axis can recruit Rad51/BRCA1, triggering homologous recombination (HR).

**Figure 4 cells-14-00748-f004:**
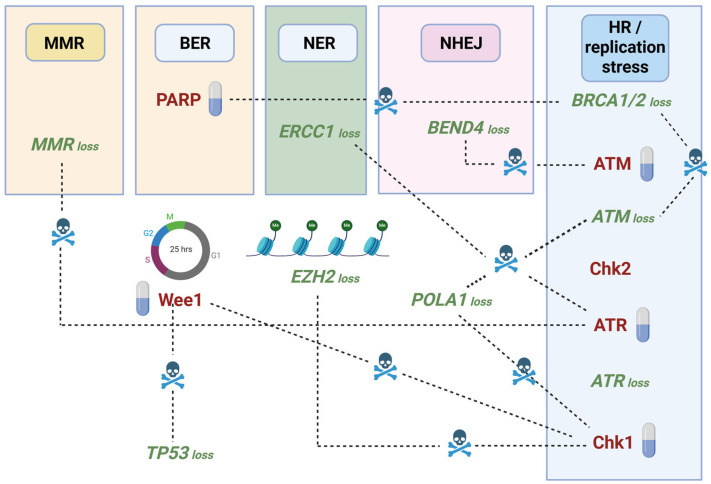
Examples of synthetic lethality due to drug inhibition or gene loss/gene silencing of ATR, Chk2, ATM, and Chk1 kinases in conjunction with other drugs or defective genes implicated in DDR, cell cycle or chromatin remodeling.

**Table 1 cells-14-00748-t001:** Selected list of inhibitors targeting ATR, AMT, Chk1, and Chk2, used alone or in combinations, in preclinical and/or clinical trials.

	Inhibitor	Treatment	Type of Cancer	Phase	Identifier
ATRi	Berzosertib (M6620, VX-970)	Cisplatin and capecitabine	Esophageal adenocarcinoma Squamous cell adenocarcinoma	I	NCT03641547
		Topotecan	Small cell lung cancer	I/II	NCT02487095
		Gemcitabine	Ovarian serous tumor Recurrent fallopian tube carcinoma Recurrent ovarian carcinoma	II	NCT02595892
		Carboplatin, gemcitabine, cisplatin, etoposide, and irinotecan	Advanced solid tumors	I	NCT02157792
		Cisplatin and gemcitabine	Urothelial carcinoma	I/II	NCT02567409
		Monotherapy	Advanced solid tumors	II	NCT03718091
	Ceralasertib (AZD6738)	Paclitaxel	Refractory cancer	I	NCT02630199
		Olaparib	High-grade serous ovarian carcinoma	I	NCT03462342
			Small cell lung cancer	II	NCT03428607
		X-ray radiotherapy	Solid tumor refractory to conventional treatment	I	NCT02223923
		Durmalumab and olaparib	Bile duct cancer	II	NCT04298021
		Monotherapy	Leukemia Myelodysplastic syndrome	I	NCT03770429
	Elimusertib (BAY 1895344)	Monotherapy	Advanced solid tumors Non-Hodgkins’ lymphoma	I	NCT03188965
			Relapsed or refractory solid tumors	I/II	NCT05071209
		Pembrolizumab	Advanced solid tumors	I	NCT04095273
		Radiation therapy and pembrolizumab	Head and neck squamous cell carcinoma	I	NCT04576091
		Leucovorin, fluorouracil, and irinotecan	Advanced or metastatic cancers of the stomach and intestines	I	NCT04535401
		Cisplatin and gemcitabine	Advanced solid tumors	I	NCT04491942
	Tuvusertib (M1774)	Niraparib	Metastatic or aocally Advanced unresectable solid tumors	I	NCT04170153
		Avelumab	ARID1 A-mutated endometrial cancer	II	NCT06518564
		Cemiplimab	Non-small cell lung cancer	I/II	NCT05882734
ATMi	AZD0156	Olaparib and irinotecan	Advanced solid tumors	I	NCT02588105
	AZD1390	Radiotherapy and durmalumav	Non-small cell lung cancer	I	NCT04550104
			Soft tissue sarcoma	I	NCT05116254
		Radiotherapy	Brain cancer	I	NCT03423628
		Stereotactic body radiotherapy	Metastatic solid tumors	I	NCT05678010
		Monotherapy	Grade 4 glioblastoma	I	NCT05182905
Chk1/2i	Prexasertib (LY-2606368)	Olaparib	Solid tumors	I	NCT03057145
		Monotherapy	Ovarian cancer Breast cancer Prostate cancer	II	NCT02203513
			Ovarian cancer	II	NCT03414047
			Neoplasms	I	NCT02514603
		Cisplatin, cetuximab, fluorouracil, LY3023414, leucovorin	Solid tumors	I	NCT02124148
		Mitoxantrone, etoposide, and cytarabine	Acute myeloid leukemia Myelodysplastic syndromes	I	NCT03735446
Chk1i	GDC-0575	Gemcitabine	Lymphoma Solid tumors	I	NCT01564251
Chk2i	PHI-101	Monotherapy	Peritoneal, fallopian or ovarian cancer	I	NCT04678102

## Data Availability

No new data were created or analyzed in this study.

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
