# Peer review of "Cancer Vulnerabilities Through Targeting the ATR/Chk1 and ATM/Chk2 Axes in the Context of DNA Damage"

_cells, 2025, doi:10.3390/cells14100748_

Round 1
Reviewer 1 Report
Comments and Suggestions for Authors
Review of the manuscript entitled
‘Cancer Vulnerabilities Through Targeting the ATR/Chk1 and 2 ATM/Chk2 Axes in the Context of DNA Damage’
by Anell Fernandez, Maider Artola, Sergio Leon, Nerea Otegui, Aroa Jimeno, Diego Serrano, and Alfonso Calvo
This review article explores the therapeutic landscape of targeting the ATR/Chk1 and ATM/Chk2 axes—two critical DNA damage response (DDR) signalling pathways. With a growing focus on personalized medicine, the authors present an analysis of how pharmacologic inhibition of these kinases offers a strategic route for enhancing cancer therapy, exploiting synthetic lethality, and potentiating immunotherapy. The review thoroughly explains how cells respond to DNA damage via repair pathways (BER, NER, MMR, HR, and NHEJ), setting the stage for understanding why cancer cells with defective repair mechanisms are selectively vulnerable. By integrating basic science with clinical insights and highlighting novel immunological implications, the article offers a strong foundation for future research and therapeutic strategies. While slightly overextended, it remains a valuable contribution to the field of precision cancer therapy.
Major concerns
#1 The review could benefit from a summary table.
#2 BER, NER, and MMR have been categorized as DNA repair pathways for single-strand DNA breaks (SSBs), as indicated in subsection 2.2.1 and Figure 1. While all three pathways involve an excision step that generates SSBs as intermediates during the repair process, SSBs are not the primary type of DNA damage these mechanisms are designed to repair. Instead, SSBs are mainly resolved by direct ligation or through PARP1-mediated repair pathways. The classification of SSBs as being repaired via the BER mechanism is, therefore, debatable. BER is widely recognized as a pathway initiated by DNA glycosylases, primarily targeting small, non-helix-distorting base lesions rather than SSBs per se. Consequently, the current categorization may require further clarification or revision within the manuscript to avoid conceptual ambiguity. Moreover, BER pathway presented on Fig. 1 should include DNA glycosylases and AP endonuclease. Additionally, NER pathway should include XPF and XPG endonucleases on Fig. 1. Furthermore, Fig. 1 displays a close resemblance to previously published data [9].
#3 Line 170 In eukaryotes, MMR strand discrimination does not rely on methylation status but rather on the presence of DNA nicks and the orientation of PCNA. Please correct.
#4 Line 181 The author states that ‘Generally, oxidation derivatives of purines and pyrimidines are the most prevalent damage produced [38].’ 8-oxoG is formed at a frequency of 1,000–2,000 lesions per cell per day, whereas damage resulting from depurination due to spontaneous hydrolysis amounts to approximately 18,000. Please correct the statement.
#5 Lines 297-298 The author states that ‘Moreover, ATM directly contributes to NHEJ and HR pathways by recruitment of repair proteins, such as DNA-PKcs or BRCA1 to the damage site [75].’ Please check the participation of CtIP and 53BP1 in the decision-making process of double-stranded DNA break repair.
#6 Lines 303 and 430 The authors write that ATM leads to cell cycle arrest at the G1/S transition. The presence of DSBs and the activation of DDR through ATM can lead to both cell cycle arrest at the G1/S and G2/M transitions. Please adjust.
#7 Figure 2 presents the activation of apoptosis through p21. The apoptosis can be activated by p53, but not p21. Please correct.
#8 Lines 361-363 The authors write that ‘As previously mentioned, ATR is deeply involved in responses against single-strand DNA damage or replication stress. Therefore, cancer cells that present these deficiencies will more likely depend on the ATR pathway for survival [88].’
These sentences can be interpreted to imply that the loss or impairment of one DNA damage response (DDR) pathway may sensitize cancer cells to treatment by increasing their reliance on the same pathway. On the other hand, cancer cells with defects in one DDR mechanism often become dependent on others for survival, and targeting these alternative/compensatory pathways can be exploited therapeutically to induce selective cancer cell death. This interpretation aligns with the rephrased explanation provided in [88]. Notably, the authors reinforce this rationale by citing publications that exemplify synthetic lethality [93, 97, 124, 128].
Minor concerns
Line 86 Instead of ‘DNA damage can be categorized in exogenous or endogenous’ it should be ‘DNA damage can be categorized as exogenous or endogenous’
Line 105 Instead of ‘ATM is mainly activated by single-stranded DNA (ssDNA) breaks’ it should be ‘ATR is mainly activated by single-stranded DNA (ssDNA) breaks’
Lines 140-141 Instead of ‘These base mismatches are repaired through mismatch repair systems [23].’ it should be ‘These base mismatches are repaired through mismatch repair system [23].’
Line 221 Reference 58 is irrelevant. It concerns γH2AX foci formation in the absence of DNA damage.
Line 315 Instead of ‘Cdc25’ there should be ‘Cdc23A’.
Line 337 Please use consequently berzosertib instead of M6620. It leads to less confusion for readers.
Line 381 Please consider rephrasing ‘additional ATR inhibitor’ into ‘other ATR inhibitor’.
Lines 387, 389, 393, 414, 420 Please add literature position at the end of the sentences.
Line 501 Please use ‘MMR-deficient’ instead of ‘mismatch repair-deficient’.
Line 505 Please consider using ‘Chk1 inhibitors’ instead of ‘Chk1 drugs’.
Line 566 Please use ‘phase I clinical trial’ instead of ‘phase 1 clinical trial’.
Line 589 Please change ‘orchestrate repair’ to ‘orchestrate DNA repair’.
Signes γ (γH2AX), δ (polδ), μ (polμ), are invisible in the text. Please verify it.
References require some adjustments: positions 18, 34 and 11 are the same publication, similarly positions 61 and 74.

Author Response
Referee 1
Comment
This review article explores the therapeutic landscape of targeting the ATR/Chk1 and ATM/Chk2 axes—two critical DNA damage response (DDR) signalling pathways. With a growing focus on personalized medicine, the authors present an analysis of how pharmacologic inhibition of these kinases offers a strategic route for enhancing cancer therapy, exploiting synthetic lethality, and potentiating immunotherapy. The review thoroughly explains how cells respond to DNA damage via repair pathways (BER, NER, MMR, HR, and NHEJ), setting the stage for understanding why cancer cells with defective repair mechanisms are selectively vulnerable. By integrating basic science with clinical insights and highlighting novel immunological implications, the article offers a strong foundation for future research and therapeutic strategies. While slightly overextended, it remains a valuable contribution to the field of precision cancer therapy.
Response
We thank the referee for the positive view about our manuscript and for the pertinent suggestions and indications, which have been all addressed.
Comment
Major concerns
#1 The review could benefit from a summary table.
Response
A Table summarizing the main different drugs, preclinical results and clinical trials has been included (Table 1).
Comment
#2 BER, NER, and MMR have been categorized as DNA repair pathways for single-strand DNA breaks (SSBs), as indicated in subsection 2.2.1 and Figure 1. While all three pathways involve an excision step that generates SSBs as intermediates during the repair process, SSBs are not the primary type of DNA damage these mechanisms are designed to repair. Instead, SSBs are mainly resolved by direct ligation or through PARP1-mediated repair pathways. The classification of SSBs as being repaired via the BER mechanism is, therefore, debatable. BER is widely recognized as a pathway initiated by DNA glycosylases, primarily targeting small, non-helix-distorting base lesions rather than SSBs per se. Consequently, the current categorization may require further clarification or revision within the manuscript to avoid conceptual ambiguity. Moreover, BER pathway presented on Fig. 1 should include DNA glycosylases and AP endonuclease. Additionally, NER pathway should include XPF and XPG endonucleases on Fig. 1. Furthermore, Fig. 1 displays a close resemblance to previously published data [9].
Response
We have taken into consideration the comments about the classification of BER, NER and MMR:
“BER, NER, and MMR have been categorized as DNA repair pathways for single-strand DNA breaks (SSBs), as indicated in subsection 2.2.1 and Figure 1. While all three pathways involve an excision step that generates SSBs as intermediates during the repair process, SSBs are not the primary type of DNA damage these mechanisms are designed to repair. Instead, SSBs are mainly resolved by direct ligation or through PARP1-mediated repair pathways.”
We have followed the widely used system of classification, but it is true that it may not be totally accurate. To clarify this issue, we have added the following sentence (lines 212-217):
“Although BER, NER, and MMR pathways involve excision steps that transiently generate SSBs as intermediates, these pathways are not primarily responsible for the repair of pre-existing SSBs. Instead, SSBs arising independently are typically resolved through direct ligation mechanisms or PARP1-dependent SSB repair pathways. Therefore, although BER, NER and MMR have been included in the SSB-repair mechanisms, this classification should take into consideration this caveat.”
In addition, Fig. 1, which has been largely changed, includes now DNA glycosylases and AP endonuclease, and NER pathway include XP-endonucleases.
Comment
#3 Line 170 In eukaryotes, MMR strand discrimination does not rely on methylation status but rather on the presence of DNA nicks and the orientation of PCNA. Please correct.
Response
Corrections have been made.
Comment
#4 Line 181 The author states that ‘Generally, oxidation derivatives of purines and pyrimidines are the most prevalent damage produced [38].’ 8-oxoG is formed at a frequency of 1,000–2,000 lesions per cell per day, whereas damage resulting from depurination due to spontaneous hydrolysis amounts to approximately 18,000. Please correct the statement.
Response
We apologize for the mistake. The sentence has been corrected.
Comment
#5 Lines 297-298 The author states that ‘Moreover, ATM directly contributes to NHEJ and HR pathways by recruitment of repair proteins, such as DNA-PKcs or BRCA1 to the damage site [75].’ Please check the participation of CtIP and 53BP1 in the decision-making process of double-stranded DNA break repair.
Response
We have added the participation of DNA-PKcs or BRCA1 to the damage site and the participation of CtIP and 53BP1 in the decision-making process of double-stranded DNA break repair (lines 328-336).
Comment
#6 Lines 303 and 430. The authors write that ATM leads to cell cycle arrest at the G1/S transition. The presence of DSBs and the activation of DDR through ATM can lead to both cell cycle arrest at the G1/S and G2/M transitions. Please adjust.
Response
We have corrected this information.
Comment
#7 Figure 2 presents the activation of apoptosis through p21. The apoptosis can be activated by p53, but not p21. Please correct.
Response
Apologies for this mistake, which has been corrected.
Comment
#8 Lines 361-363 The authors write that ‘As previously mentioned, ATR is deeply involved in responses against single-strand DNA damage or replication stress. Therefore, cancer cells that present these deficiencies will more likely depend on the ATR pathway for survival [88].’
These sentences can be interpreted to imply that the loss or impairment of one DNA damage response (DDR) pathway may sensitize cancer cells to treatment by increasing their reliance on the same pathway. On the other hand, cancer cells with defects in one DDR mechanism often become dependent on others for survival, and targeting these alternative/compensatory pathways can be exploited therapeutically to induce selective cancer cell death. This interpretation aligns with the rephrased explanation provided in [88]. Notably, the authors reinforce this rationale by citing publications that exemplify synthetic lethality [93, 97, 124, 128].
Response
We have removed this sentence (“Therefore, cancer cells that present these deficiencies will more likely depend on the ATR pathway for survival”) to avoid any possibility of confusion.
Comment
Minor concerns
Line 86 Instead of ‘DNA damage can be categorized in exogenous or endogenous’ it should be ‘DNA damage can be categorized as exogenous or endogenous’
Line 105 Instead of ‘ATM is mainly activated by single-stranded DNA (ssDNA) breaks’ it should be ‘ATR is mainly activated by single-stranded DNA (ssDNA) breaks’
Lines 140-141 Instead of ‘These base mismatches are repaired through mismatch repair systems [23].’ it should be ‘These base mismatches are repaired through mismatch repair system [23].’
Line 221 Reference 58 is irrelevant. It concerns γH2AX foci formation in the absence of DNA damage.
Line 315 Instead of ‘Cdc25’ there should be ‘Cdc23A’.
Line 337 Please use consequently berzosertib instead of M6620. It leads to less confusion for readers.
Line 381 Please consider rephrasing ‘additional ATR inhibitor’ into ‘other ATR inhibitor’.
Lines 387, 389, 393, 414, 420 Please add literature position at the end of the sentences.
Line 501 Please use ‘MMR-deficient’ instead of ‘mismatch repair-deficient’.
Line 505 Please consider using ‘Chk1 inhibitors’ instead of ‘Chk1 drugs’.
Line 566 Please use ‘phase I clinical trial’ instead of ‘phase 1 clinical trial’.
Line 589 Please change ‘orchestrate repair’ to ‘orchestrate DNA repair’.
Signes γ (γH2AX), δ (polδ), μ (polμ) are invisible in the text. Please verify it.
References require some adjustments: positions 18, 34 and 11 are the same publication, similarly positions 61 and 74.
Response
All the minor issues have been revised and modified according to the comments.
Reviewer 2 Report
Comments and Suggestions for Authors
The attached file

Author Response
Referee 2
Comment
Report on manuscript: cells-3582624 Title: Cancer vulnerabilities through targeting the ATR/Chk1 and 2 ATM/Chk2 axes in the context of DNA damage
Authors: Fernandez et al.
The authors review the impact of targeting DNA damage response pathways, in particular the ATM/Chk2 and ATR/Chk1 axes, in combination with new drugs developed to inhibit these proteins to widen the therapeutic window. In addition to basic information on the respective pathways, examples of pre-clinical or clinical studies are presented for the different types of damage induced by radiotherapy and chemotherapy. The focus of the review is on recent discoveries involving the targeting of these proteins in specific tumour genetic backgrounds associated with DNA damage, the inhibition of which leads to synthetic lethality. Finally, the role of ATM/Chk2 and ATR/Chk1 blockade as a means of enhancing the response to immunotherapy is discussed. In general, this review provides a good summary of the scientific state of the art in exploiting the vulnerability of different cancer cell types by targeting the ATR/Chk1 and 2 ATM/Chk2 axes in the context of DNA damage.
Main comments on this paper:
Point 1: Although the language is good, it is still difficult to understand some paragraphs. Grammatical and semantic errors should be corrected.
Response
We thank the referee for these comments. Language has been carefully reviewed.
Comment
Point 2: Page 2, lines 83-88: The authors describe the activation of ATM and ATR. There seems to be a spelling error. ATM is mainly activated by double-stranded DNA, not single-stranded DNA, whereas ATR is mainly activated by single-stranded DNA.
Response
We apologize for this mistake, which has been corrected.
Comment
Point 3: Page 6, lines 220-222: There seems to be a typo in the following sentence, the symbol for gamma is missing. It should read: "These proteins play a crucial role in the phosphorylation of histone H2AX at Ser139, converting it into gamma-H2AX...".
Response
We have edited the text, so that g-H2AX is now included.
Comment
Point 4: Page 6, 2nd paragraph: In addition to the well-established non-homologous end joining (NHEJ) process, it is imperative to acknowledge the role of the alternative pathway in order to ensure the comprehensive coverage of the most pertinent pathways involved in the DNA damage response of cells.
Response
We have added a text explaining the alternative NHEJ pathway (see lines 237-249).
Comment
Point 5: A tabular summary of the inhibitors for the ATR/CHk1 and ATM/Chk2 axis, along with a compendium of preclinical and clinical studies published on these, would prove beneficial.
Response
A tabular summary of the inhibitors for the ATR/Chk1 and ATM/Chk2 axes, along with a compendium of preclinical and clinical studies published has been added in the manuscript (Table 1).
Reviewer 3 Report
Comments and Suggestions for Authors
This manuscript presents a comprehensive review of the ATR/Chk1 and ATM/Chk2 signaling pathways within the DNA damage response (DDR) and explores their therapeutic targeting in cancer. The authors discuss the mechanistic basis of DDR, highlight synthetic lethality strategies, and detail the progress of clinical trials involving small-molecule inhibitors targeting these pathways. Additionally, the manuscript addresses the synergy between DDR inhibition and immunotherapy, making it highly relevant to current cancer treatment paradigms. Overall, the manuscript is logically organized, transitioning effectively from molecular mechanisms to therapeutic applications, and is supported by informative figures and thorough citations. Below are some suggestions that should be addressed prior to acceptance of the manuscript.
- The manuscript would benefit from a deeper analysis of challenges in clinical translation, including: 1) Resistance mechanisms to DDR inhibitors 2) Toxicity concerns in combinatorial therapies 3) Limitations in biomarker-driven patient stratification.
- While ATR/Chk1 inhibitors are extensively covered, the ATM/Chk2 axis requires a more balanced mechanistic discussion and greater emphasis on therapeutic strategies.
- A schematic diagram illustrating the synthetic lethality targeting of the ATR/Chk1 and ATM/Chk2 pathways should be included.
- A schematic diagram for section 6 is also should be provided to improve readability.
- A visual summary of the DDR pathways and their therapeutic modulation—including synthetic lethality and immunotherapy synergy—would significantly enhance reader comprehension and the overall impact of the article.
- Some redundancy is present in the sections describing DNA repair pathways (MMR, BER, HR). Consider condensing these parts to maintain focus and improve readability.
- Minor grammatical and stylistic issues are noted. A final round of proofreading by a native English speaker is recommended to improve fluency and clarity.
Minor grammatical and stylistic issues are noted. A final round of proofreading by a native English speaker is recommended to improve fluency and clarity.
Author Response
Referee 3
Comment
This manuscript presents a comprehensive review of the ATR/Chk1 and ATM/Chk2 signaling pathways within the DNA damage response (DDR) and explores their therapeutic targeting in cancer. The authors discuss the mechanistic basis of DDR, highlight synthetic lethality strategies, and detail the progress of clinical trials involving small-molecule inhibitors targeting these pathways. Additionally, the manuscript addresses the synergy between DDR inhibition and immunotherapy, making it highly relevant to current cancer treatment paradigms. Overall, the manuscript is logically organized, transitioning effectively from molecular mechanisms to therapeutic applications, and is supported by informative figures and thorough citations. Below are some suggestions that should be addressed prior to acceptance of the manuscript.
Response
We thank the referee for the comments and suggestions. A Table showing relevant inhibitors for the ATR/CHk1 and ATM/Chk2 axis has been added in the manuscript (Table 1).
Comment
The manuscript would benefit from a deeper analysis of challenges in clinical translation, including: 1) Resistance mechanisms to DDR inhibitors 2) Toxicity concerns in combinatorial therapies 3) Limitations in biomarker-driven patient stratification.
Response
Thank you for this suggestion, which we acknowledge also as an important issue. We have now added a text dealing with all these issues (lines 679-689).
Comment
While ATR/Chk1 inhibitors are extensively covered, the ATM/Chk2 axis requires a more balanced mechanistic discussion and greater emphasis on therapeutic strategies.
Response
We have added more information about the AMT/Chk2 axis regarding therapeutic strategies.
Comment
A schematic diagram illustrating the synthetic lethality targeting of the ATR/Chk1 and ATM/Chk2 pathways should be included.
Response
Following the referee’s indication, a schematic diagram showing relevant synthetic lethality targeting the ATR/Chk1 and ATM/Chk2 axes together with other pathways, has been included.
Comment
A schematic diagram for section 6 is also should be provided to improve readability.
A visual summary of the DDR pathways and their therapeutic modulation—including synthetic lethality and immunotherapy synergy—would significantly enhance reader comprehension and the overall impact of the article.
Response
We have increased the number of figures from two to four and added a table (Table 1). While we acknowledge that every section could benefit from a diagram, we believe the figures we have included clarify the most essential parts of the text.
Comment
Some redundancy is present in the sections describing DNA repair pathways (MMR, BER, HR). Consider condensing these parts to maintain focus and improve readability.
Minor grammatical and stylistic issues are noted. A final round of proofreading by a native English speaker is recommended to improve fluency and clarity.
Minor grammatical and stylistic issues are noted. A final round of proofreading by a native English speaker is recommended to improve fluency and clarity.
Response
The sections on MMR, BER, and HR have been extensively revised in accordance with the suggestions from the other two reviewers. We believe the content is now clearer and easier to follow. The English has also been thoroughly revised.
Round 2
Reviewer 1 Report
Comments and Suggestions for Authors
Kindly include in the text the following statement, which was provided in the authors’ response to reviewer’s comment #2.
'Although BER, NER, and MMR pathways involve excision steps that transiently generate SSBs as intermediates, these pathways are not primarily responsible for the repair of pre-existing SSBs. Instead, SSBs arising independently are typically resolved through direct ligation mechanisms or PARP1-dependent SSB repair pathways. Therefore, although BER, NER and MMR have been included in the SSB-repair mechanisms, this classification should take into consideration this caveat'
Author Response
Pamplona, 08 May 2025
Editorial board CELLS
Manuscript ID: cells-3582624
Dear Editor:
We have added the required sentence that was not included by mistake.
We hope that our study can be now accepted for publication in CELLS.
Kind regards,
Alfonso Calvo
Referee 2
Comment
“BER, NER, and MMR have been categorized as DNA repair pathways for single-strand DNA breaks (SSBs), as indicated in subsection 2.2.1 and Figure 1. While all three pathways involve an excision step that generates SSBs as intermediates during the repair process, SSBs are not the primary type of DNA damage these mechanisms are designed to repair. Instead, SSBs are mainly resolved by direct ligation or through PARP1-mediated repair pathways.”
We have followed the widely used system of classification, but it is true that it may not be totally accurate. To clarify this issue, we have added the following sentence (lines 212-217):
“Although BER, NER, and MMR pathways involve excision steps that transiently generate SSBs as intermediates, these pathways are not primarily responsible for the repair of pre-existing SSBs. Instead, SSBs arising independently are typically resolved through direct ligation mechanisms or PARP1-dependent SSB repair pathways. Therefore, although BER, NER and MMR have been included in the SSB-repair mechanisms, this classification should take into consideration this caveat.”
